# Improving Diffusion-Based Image Synthesis with Context Prediction

**Ling Yang**[1]* **Jingwei Liu**[1]† **Shenda Hong**[1] **Zhilong Zhang**[1] **Zhilin Huang**[2]
**Zheming Cai**[1] **Wentao Zhang**[1] **Bin Cui**[1] ‡
[1]Peking University [2] Tsinghua University
`yangling0818@163.com, jingweiliu1996@163.com, zhilong.zhang@bjmu.edu.cn`
`{hongshenda, wentao.zhang, bin.cui}@pku.edu.cn`

## Abstract

Diffusion models are a new class of generative models, and have dramatically promoted image generation with unprecedented quality and diversity. Existing diffusion models mainly try to reconstruct input image from a corrupted one with a pixel-wise or feature-wise constraint along spatial axes. However, such point-based reconstruction may fail to make each predicted pixel/feature fully preserve its neighborhood context, impairing diffusion-based image synthesis. As a powerful source of automatic supervisory signal, *context* has been well studied for learning representations. Inspired by this, we for the first time propose CONPREDIFF to improve diffusion-based image synthesis with context prediction. We explicitly reinforce each point to predict its neighborhood context (*i.e.*, multi-stride features/tokens/pixels) with a context decoder at the end of diffusion denoising blocks in training stage, and remove the decoder for inference. In this way, each point can better reconstruct itself by preserving its semantic connections with neighborhood context. This new paradigm of CONPREDIFF can generalize to arbitrary discrete and continuous diffusion backbones without introducing extra parameters in sampling procedure. Extensive experiments are conducted on unconditional image generation, text-to-image generation and image inpainting tasks. Our CONPREDIFF consistently outperforms previous methods and achieves a new SOTA text-to-image generation results on MS-COCO, with a zero-shot FID score of 6.21.

## 1 Introduction

Recent diffusion models [99, 5, 63, 4, 47, 10, 22] have made remarkable progress in image generation. They are first introduced by Sohl-Dickstein et al. [76] and then improved by Song & Ermon [78] and Ho et al. [28], and can now generate image samples with unprecedented quality and diversity [24, 68, 67]. Numerous methods have been proposed to develop diffusion models by improving their empirical generation results [53, 77, 79] or extending the capacity of diffusion models from a theoretical perspective [80, 81, 47, 46, 108]. We revisit existing diffusion models for image generation and break them into two categories, pixel- and latent-based diffusion models, according to their diffusing spaces. Pixel-based diffusion models directly conduct continuous diffusion process in the pixel space, they incorporate various conditions (*e.g.*, class, text, image, and semantic map) [29, 70, 51, 2, 66] or auxiliary classifiers [80, 14, 27, 54, 40] for conditional image generation.

On the other hand, latent-based diffusion models [65] conduct continuous or discrete diffusion process [87, 30, 1] on the semantic latent space. Such diffusion paradigm not only significantly reduces the

---

*Contact: Ling Yang, yangling0818@163.com.

†Contributed equally.

‡Corresponding Authors: Wentao Zhang, Bin Cui.

computational complexity for both training and inference, but also facilitates the conditional image generation in complex semantic space [62, 38, 58, 19, 98]. Some of them choose to pre-train an autoencoder [41, 64] to map the input from image space to the continuous latent space for continuous diffusion, while others utilize a vector quantized variational autoencoder [88, 17] to induce the token-based latent space for discrete diffusion [24, 75, 114, 85].

Despite all these progress of pixel- and latent-based diffusion models in image generation, both of them mainly focus on utilizing a point-based reconstruction objective over the spatial axes to recover the entire image in diffusion training process. This point-wise reconstruction neglects to fully preserve local context and semantic distribution of each predicted pixel/feature, which may impair the fidelity of generated images. Traditional non-diffusion studies [15, 45, 32, 50, 110, 8] have designed different *context*-preserving terms for advancing image representation learning, but few researches have been done to constrain on context for diffusion-based image synthesis.

In this paper, we propose CONPREDIFF to explicitly force each pixel/feature/token to predict its local neighborhood context (*i.e.*, multi-stride features/tokens/pixels) in image diffusion generation with an extra context decoder near the end of diffusion denoising blocks. This explicit *context prediction* can be extended to existing discrete and continuous diffusion backbones without introducing additional parameters in inference stage. We further characterize the neighborhood context as a probability distribution defined over multi-stride neighbors for efficiently decoding large context, and adopt an optimal-transport loss based on Wasserstein distance [21] to impose structural constraint between the decoded distribution and the ground truth. We evaluate the proposed CONPREDIFF with the extensive experiments on three major visual tasks, unconditional image generation, text-to-image generation, and image inpainting. Notably, our CONPREDIFF consistently outperforms previous diffusion models by a large margin regarding generation quality and diversity.

Our main contributions are summarized as follows: **(i):** To the best of our knowledge, we for the first time propose CONPREDIFF to improve diffusion-based image generation with context prediction; **(ii):** We further propose an efficient approach to decode large context with an optimal-transport loss based on Wasserstein distance; **(iii):** CONPREDIFF substantially outperforms existing diffusion models and achieves new SOTA image generation results, and we can generalize our model to existing discrete and continuous diffusion backbones, consistently improving their performance.

## 2  Related Work

**Diffusion Models for Image Generation**    Diffusion models [99, 76, 78, 28] are a new class of probabilistic generative models that progressively destruct data by injecting noise, then learn to reverse this process for sample generation. They can generate image samples with unprecedented quality and diversity [24, 68, 67], and have been applied in various applications [99, 9, 6]. Existing pixel- and latent-based diffusion models mainly utilize the discrete diffusion [30, 1, 24] or continuous diffusion [87, 65] for unconditional or conditional image generation [80, 14, 27, 54, 40, 68]. Discrete diffusion models were also first described in [76], and then applied to text generation in Argmax Flow [30]. D3PMs [1] applies discrete diffusion to image generation. VQ-Diffusion [24] moves discrete diffusion from image pixel space to latent space with the discrete image tokens acquired from VQ-VAE [88]. Latent Diffusion Models (LDMs) [87, 65] reduce the training cost for high resolution images by conducting continuous diffusion process in a low-dimensional latent space. They also incorporate conditional information into the sampling process via cross attention [89]. Similar techniques are employed in DALLE-2 [62] for image generation from text, where the continuous diffusion model is conditioned on text embeddings obtained from CLIP latent codes [59]. Imagen [68] implements text-to-image generation by conditioning on text embeddings acquired from large language models (*e.g.*, T5 [60]). Despite all this progress, existing diffusion models neglect to exploit rich neighborhood context in the generation process, which is critical in many vision tasks for maintaining the local semantic continuity in image representations [111, 45, 32, 50]. In this paper, we firstly propose to explicitly preserve local neighborhood context for diffusion-based image generation.

**Context-Enriched Representation Learning**    *Context* has been well studied in learning representations, and is widely proved to be a powerful automatic supervisory signal in many tasks. For example, language models [52, 13] learn word embeddings by predicting their context, *i.e.*, a few words before and/or after. More utilization of contextual information happens in visual tasks, where spatial context

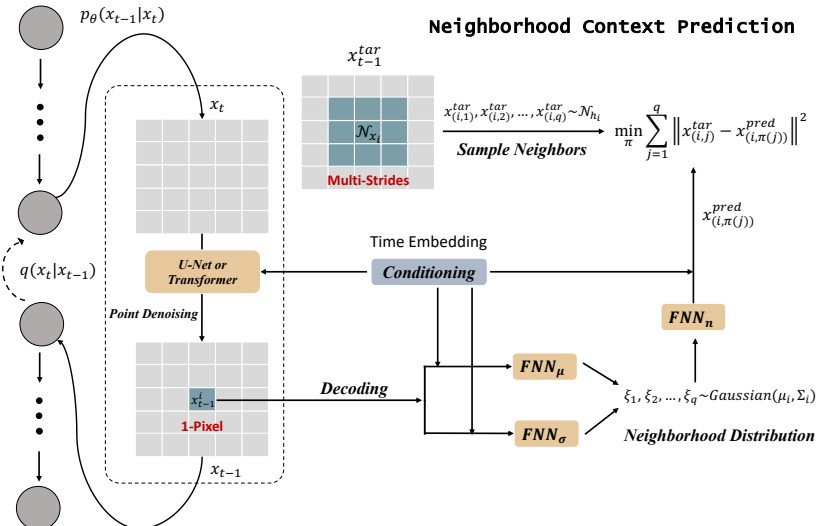

Figure 1: In training stage, CONPREDIFF first performs self-denoising as standard diffusion models, then it conducts neighborhood context prediction based on denoised point $x_{t-1}^i$. In inference stage, CONPREDIFF only uses its self-denoising network for sampling.

is vital for image domain. Many studies [15, 111, 45, 32, 50, 110, 8, 106, 95, 94, 44] propose to leverage context for enriching learned image representations. Doersch et al. [15] and Zhang et al. [110] make predictions from visible patches to masked patches to enhance the self-supervised image representation learning. Hu et al. [32] designs local relation layer to model the context of local pixel pairs for image classification, while Liu et al. [45] preserves contextual structure to guarantee the local feature/pixel continuity for image inpainting. Inspired by these studies, in this work, we propose to incorporate neighborhood context prediction for improving diffusion-based generative modeling.

## 3 Preliminary

**Discrete Diffusion**    We briefly review a classical discrete diffusion model, namely Vector Quantized Diffusion (VQ-Diffusion) [24]. VQ-Diffusion utilizes a VQ-VAE to convert images $x$ to discrete tokens $x_0 \in \{1, 2, ..., K, K + 1\}$, $K$ is the size of codebook, and $K + 1$ denotes the [MASK] token. Then the forward process of VQ-Diffusion is given by:

$$q(x_t|x_{t-1}) = \boldsymbol{v}^\top(x_t)\boldsymbol{Q}_t\boldsymbol{v}(x_{t-1}) \tag{1}$$

where $\boldsymbol{v}(x)$ is a one-hot column vector with entry 1 at index $x$. And $\boldsymbol{Q}_t$ is the probability transition matrix from $x_{t-1}$ to $x_t$ with the mask-and-replace VQ-Diffusion strategy. In the reverse process, VQ-Diffusion trains a denoising network $p_\theta(\boldsymbol{x}_{t-1}|\boldsymbol{x}_t)$ that predicts noiseless token distribution $p_\theta(\tilde{\boldsymbol{x}}_0|\boldsymbol{x}_t)$ at each step:

$$p_\theta(\boldsymbol{x}_{t-1}|\boldsymbol{x}_t) = \sum_{\tilde{\boldsymbol{x}}_0=1}^{K} q(\boldsymbol{x}_{t-1}|\boldsymbol{x}_t, \tilde{\boldsymbol{x}}_0)p_\theta(\tilde{\boldsymbol{x}}_0|\boldsymbol{x}_t), \tag{2}$$

which is optimized by minimizing the following variational lower bound (VLB) [76]:

$$\mathcal{L}_{t-1}^{dis} = D_{KL}(q(\boldsymbol{x}_{t-1}|\boldsymbol{x}_t, \boldsymbol{x}_0) \,||\, p_\theta(\boldsymbol{x}_{t-1}|\boldsymbol{x}_t)). \tag{3}$$

**Continuous Diffusion**    A continuous diffusion model progressively perturbs input image or feature map $\boldsymbol{x}_0$ by injecting noise, then learn to reverse this process starting from $\boldsymbol{x}_T$ for image generation. The forward process can be formulated as a Gaussian process with Markovian structure:

$$\begin{aligned} q(\boldsymbol{x}_t|\boldsymbol{x}_{t-1}) &:= \mathcal{N}(\boldsymbol{x}_t; \sqrt{1 - \beta_t}\boldsymbol{x}_{t-1}, \beta_t\mathbf{I}), \\ q(\boldsymbol{x}_t|\boldsymbol{x}_0) &:= \mathcal{N}(\boldsymbol{x}_t; \sqrt{\overline{\alpha}_t}\boldsymbol{x}_0, (1 - \overline{\alpha}_t)\mathbf{I}), \end{aligned} \tag{4}$$

where $\beta_1, \ldots, \beta_T$ denotes fixed variance schedule with $\alpha_t := 1 - \beta_t$ and $\overline{\alpha}_t := \prod_{s=1}^{t} \alpha_s$. This forward process progressively injects noise to data until all structures are lost, which is well approximated by $\mathcal{N}(0, \mathbf{I})$. The reverse diffusion process learns a model $p_\theta(\boldsymbol{x}_{t-1}|\boldsymbol{x}_t)$ that approximates the

true posterior:

$$p_\theta(\boldsymbol{x}_{t-1}|\boldsymbol{x}_t) := \mathcal{N}(\boldsymbol{x}_{t-1}; \mu_\theta(\boldsymbol{x}_t), \Sigma_\theta(\boldsymbol{x}_t)), \tag{5}$$

Fixing $\Sigma_\theta$ to be untrained time dependent constants $\sigma_t^2 \boldsymbol{I}$, Ho *et al.* [28] improve the diffusion training process by optimizing following objective:

$$\mathcal{L}_{t-1}^{con} = \underset{q(\boldsymbol{x}_t|\boldsymbol{x}_{t-1})}{\mathbb{E}} \left[ \frac{1}{2\sigma_t^2} ||\mu_\theta(\boldsymbol{x}_t, t) - \hat{\mu}(\boldsymbol{x}_t, \boldsymbol{x}_0)||^2 \right] + C, \tag{6}$$

where $C$ is a constant that does not depend on $\theta$. $\hat{\mu}(\boldsymbol{x}_t, \boldsymbol{x}_0)$ is the mean of the posterior $q(\boldsymbol{x}_{t-1}|\boldsymbol{x}_0, \boldsymbol{x}_t)$, and $\mu_\theta(\boldsymbol{x}_t, t)$ is the predicted mean of $p_\theta(\boldsymbol{x}_{t-1} \mid \boldsymbol{x}_t)$ computed by neural networks.

## 4 The Proposed CONPREDIFF

In this section, we elucidate the proposed CONPREDIFF as in Figure 1. In Sec. 4.1, we introduce our proposed context prediction term for explicitly preserving local neighborhood context in diffusion-based image generation. To efficiently decode large context in training process, we characterize the neighborhood information as the probability distribution defined over multi-stride neighbors in Sec. 4.2, and theoretically derive an optimal-transport loss function based on Wasserstein distance to optimize the decoding procedure. In Sec. 4.3, we generalize our CONPREDIFF to both existing discrete and continuous diffusion models, and provide optimization objectives.

### 4.1 Neighborhood Context Prediction in Diffusion Generation

We use unconditional image generation to illustrate our method for simplicity. Let $\boldsymbol{x}_{t-1}^i \in \mathbb{R}^d$ to denote $i$-th pixel of the predicted image, $i$-th feature point of the predicted feature map, or $i$-th image token of the predicted token map in spatial axes. Let $\mathcal{N}_i^s$ denote the $s$-stride neighborhoods of $\boldsymbol{x}_{t-1}^i$, and $K$ denotes the total number of $\mathcal{N}_i^s$. For example, the number of 1-stride neighborhoods is $K = 8$, and the number of 2-stride ones is $K = 24$.

$S$-**Stride Neighborhood Reconstruction** Previous diffusion models make point-wise reconstruction, *i.e.*, reconstructing each pixel, thus their reverse learning processes can be formulated by $p_\theta(\boldsymbol{x}_{t-1}^i|\boldsymbol{x}_t)$. In contrast, our context prediction aims to reconstruct $\boldsymbol{x}_{t-1}^i$ and further predict its $s$-stride neighborhood contextual representations $\boldsymbol{H}_{\mathcal{N}_i^s}$ based on $\boldsymbol{x}_{t-1}^i$: $p_\theta(\boldsymbol{x}_{t-1}^i, \boldsymbol{H}_{\mathcal{N}_i^s}|\boldsymbol{x}_t)$, where $p_\theta$ is parameterized by two reconstruction networks ($\psi_p, \psi_n$). $\psi_p$ is designed for the point-wise denoising of $\boldsymbol{x}_{t-1}^i$ in $\boldsymbol{x}_t$, and $\psi_n$ is designed for decoding $\boldsymbol{H}_{\mathcal{N}_i^s}$ from $\boldsymbol{x}_{t-1}^i$. For denoising $i$-th point in $\boldsymbol{x}_t$, we have:

$$\boldsymbol{x}_{t-1}^i = \psi_p(\boldsymbol{x}_t, t), \tag{7}$$

where $t$ is the time embedding and $\psi_p$ is parameterized by a U-Net or transformer with an encoder-decoder architecture. For reconstructing the entire neighborhood information $\boldsymbol{H}_{\mathcal{N}_i^s}$ around each point $\boldsymbol{x}_{t-1}^i$, we have:

$$\boldsymbol{H}_{\mathcal{N}_i^s} = \psi_n(\boldsymbol{x}_{t-1}^i, t) = \psi_n(\psi_p(\boldsymbol{x}_t, t)), \tag{8}$$

where $\psi_n \in \mathbb{R}^{Kd}$ is the neighborhood decoder. Based on Equation (7) and Equation (8), we unify the point- and neighborhood-based reconstruction to form the overall training objective:

$$\mathcal{L}_{\text{CONPREDIFF}} = \sum_{i=1}^{x \times y} \left[ \underbrace{\mathcal{M}_p(\boldsymbol{x}_{t-1}^i, \hat{\boldsymbol{x}}^i)}_{point\ denoising} + \underbrace{\mathcal{M}_n(\boldsymbol{H}_{\mathcal{N}_i^s}, \hat{\boldsymbol{H}}_{\mathcal{N}_i^s})}_{context\ prediction} \right], \tag{9}$$

where $x, y$ are the width and height on spatial axes. $\hat{\boldsymbol{x}}^i$ ($\hat{\boldsymbol{x}}_0^i$) and $\hat{\boldsymbol{H}}_{\mathcal{N}_i^s}$ are ground truths. $\mathcal{M}_p$ and $\mathcal{M}_n$ can be Euclidean distance. In this way, CONPREDIFF is able to maximally preserve local context for better reconstructing each pixel/feature/token.

**Interpreting Context Prediction in Maximizing ELBO** We let $\mathcal{M}_p, \mathcal{M}_n$ be square loss, $\mathcal{M}_n(\boldsymbol{H}_{\mathcal{N}_i^s}, \hat{\boldsymbol{H}}_{\mathcal{N}_i^s}) = \sum_{j \in \mathcal{N}_i} (\boldsymbol{x}_0^{i,j} - \hat{\boldsymbol{x}}_0^{i,j})^2$, where $\hat{\boldsymbol{x}}_0^{i,j}$ is the j-th neighbor in the context of $\hat{\boldsymbol{x}}_0^i$ and $\boldsymbol{x}_0^{i,j}$ is the prediction of $\boldsymbol{x}_0^{i,j}$ from a denoising neural network. Thus we have:

$$\boldsymbol{x}_0^{i,j} = \psi_n(\psi_p(\boldsymbol{x}_t, t)(i))(j). \tag{10}$$

Compactly, we can write the denoising network as:

$$\Psi(x_t, t)(i, j) = \begin{cases} \psi_n(\psi_p(\boldsymbol{x}_t, t)(i))(j), & j \in \mathcal{N}_i, \\ \psi_p(\boldsymbol{x}_t, t)(i), & j = i. \end{cases} \tag{11}$$

We will show that the DDPM loss is upper bounded by ConPreDiff loss, by reparameterizing $\boldsymbol{x}_0(\boldsymbol{x}_t, t)$. Specifically, for each unit $i$ in the feature map, we use the mean of predicted value in its neighborhood as the final prediction:

$$\boldsymbol{x}_0(\boldsymbol{x}_t, t)(i) = 1/(|\mathcal{N}_i| + 1) * \sum_{j \in \mathcal{N}_i \cup \{i\}} \Psi(\boldsymbol{x}_t, t)(i, j). \tag{12}$$

Now we can show the connection between the DDPM loss and ConPreDiff loss:

$$
\begin{aligned}
||\hat{\boldsymbol{x}}_0 - \boldsymbol{x}_0(\boldsymbol{x}_t, t)||_2^2 &= \sum_i (\hat{\boldsymbol{x}}_0^i - \boldsymbol{x}_0(\boldsymbol{x}_t, t)(i))^2, \\
&= \sum_i (\hat{\boldsymbol{x}}_0^i - \sum_{j \in \mathcal{N}_i \cup \{i\}} \Psi(\boldsymbol{x}_t, t)(i, j)/(|\mathcal{N}_i| + 1))^2, \\
&= \sum_i (\sum_{j \in \mathcal{N}_i \cup \{i\}} (\Psi(\boldsymbol{x}_t, t)(i, j) - \hat{\boldsymbol{x}}_0^i))^2/(|\mathcal{N}_i| + 1)^2, \\
\text{(Cauchy Inequality)} &\leq \sum_i \sum_{j \in \mathcal{N}_i \cup \{i\}} (\Psi(\boldsymbol{x}_t, t)(i, j) - \hat{\boldsymbol{x}}_0^i)^2/(|\mathcal{N}_i| + 1), \\
&= 1/(|\mathcal{N}_i| + 1) \sum_i [(\hat{\boldsymbol{x}}_0^i - \psi_p(\boldsymbol{x}_t, t)(i))^2 + \sum_{j \in \mathcal{N}_i} (\hat{\boldsymbol{x}}_0^{i,j} - \boldsymbol{x}_0^{i,j})^2]
\end{aligned}
\tag{13}
$$

In the last equality, we assume that the feature is padded so that each unit $i$ has the same number of neighbors $|\mathcal{N}|$. As a result, the ConPreDiff loss is an upper bound of the negative log likelihood.

**Complexity Problem** We note that directly optimizing the Equation (9) has a complexity problem and it will substantially lower the efficiency of CONPREDIFF in training stage. Because the network $\psi_n : \mathbb{R}^d \to \mathbb{R}^{Kd}$ in Equation (8) needs to expand the channel dimension by $K$ times for large-context neighborhood reconstruction, it significantly increases the parameter complexity of the model. Hence, we seek for another way that is efficient for reconstructing neighborhood information.

We solve the challenging problem by changing the direct prediction of entire neighborhoods to the prediction of neighborhood distribution. Specifically, for each $\boldsymbol{x}_{t-1}^i$, the neighborhood information is represented as an empirical realization of i.i.d. sampling $Q$ elements from $\mathcal{P}_{\mathcal{N}_i^s}$, where $\mathcal{P}_{\mathcal{N}_i^s} \triangleq \frac{1}{K} \sum_{u \in \mathcal{N}_i^s} \delta_{h_u}$. Based on this view, we are able to transform the neighborhood prediction $\mathcal{M}_n$ into the neighborhood distribution prediction. **However, such sampling-based measurement loses original spatial orders of neighborhoods, and thus we use a permutation invariant loss (Wasserstein distance) for optimization**. Wasserstein distance [23, 21] is an effective metric for measuring structural similarity between distributions, which is especially suitable for our neighborhood distribution prediction. And we rewrite the Equation (9) as:

$$\mathcal{L}_{\text{CONPREDIFF}} = \sum_{i=1}^{x \times y} \left[ \underbrace{\mathcal{M}_p(\boldsymbol{x}_{t-1}^i, \hat{\boldsymbol{x}}^i)}_{point\ denoising} + \underbrace{\mathcal{W}_2^2(\psi_n(\boldsymbol{x}_{t-1}^i, t), \mathcal{P}_{\mathcal{N}_i^s})}_{neighborhood\ distribution\ prediction} \right], \tag{14}$$

where $\psi_n(\boldsymbol{x}_{t-1}^i, t)$ is designed to decode neighborhood distribution parameterized by feedforward neural networks (FNNs), and $\mathcal{W}_2(\cdot, \cdot)$ is the 2-Wasserstein distance as defined below. We provide a more explicit formulation of $\mathcal{W}_2^2(\psi_n(\boldsymbol{x}_{t-1}^i, t), \mathcal{P}_{\mathcal{N}_i^s})$ in Sec. 4.2.

**Definition 4.1.** Let $\mathcal{P}, \mathcal{Q}$ denote two probability distributions with finite second moment defined on $\mathcal{Z} \subseteq \mathbb{R}^m$. The 2-Wasserstein distance between $\mathcal{P}$ and $\mathcal{Q}$ defined on $\mathcal{Z}, \mathcal{Z}' \subseteq \mathbb{R}^m$ is the solution to the optimal mass transportation problem with $\ell_2$ transport cost [90]:

$$\mathcal{W}_2(\mathcal{P}, \mathcal{Q}) = \left( \inf_{\gamma \in \Gamma(\mathcal{P}, \mathcal{Q})} \int_{\mathcal{Z} \times \mathcal{Z}'} ||Z - Z'||_2^2 d\gamma(Z, Z') \right)^{1/2} \tag{15}$$

where $\Gamma(\mathcal{P}, \mathcal{Q})$ contains all joint distributions of $(Z, Z')$ with marginals $\mathcal{P}$ and $\mathcal{Q}$ respectively.

## 4.2 Efficient Large Context Decoding

Our CONPREDIFF essentially represents the node neighborhood $\hat{\boldsymbol{H}}_{\mathcal{N}_i^s}$ as a distribution of neighbors' representations $\mathcal{P}_{\mathcal{N}_i^s}$ (Equation (14)). We adopt Wasserstein distance to characterize the distribution reconstruction loss because $\mathcal{P}_{\mathcal{N}_i^s}$ has atomic non-zero measure supports in a continuous space, where the family of $f$-divergences such as KL-divergence cannot be applied. Maximum mean discrepancy may be applied but it needs to choose a specific kernel function.

We define the decoded distribution $\psi_n(\boldsymbol{x}_{t-1}^i, t)$ as an FNN-based transformation of a Gaussian distribution parameterized by $\boldsymbol{x}_{t-1}^i$ and $t$. The reason for choosing this setting stems from the fact that the universal approximation capability of FNNs allows to (approximately) reconstruct any distributions in 1-Wasserstein distance, as formally stated in Theorem 4.2, proved in Lu & Lu [48]. To enhance the empirical performance, our case adopts the 2-Wasserstein distance and an FNN with $d$-dim output instead of the gradient of an FNN with 1-dim outout. Here, the reparameterization trick [42] needs to be used:

$$
\begin{aligned}
\psi_n(\boldsymbol{x}_{t-1}^i, t) &= \text{FNN}_n(\xi), \ \xi \sim \mathcal{N}(\mu_i, \Sigma_i), \\
\mu_i &= \text{FNN}_\mu(\boldsymbol{x}_{t-1}^i), \Sigma_i = \text{diag}(\exp(\text{FNN}_\sigma(\boldsymbol{x}_{t-1}^i))).
\end{aligned}
\tag{16}
$$

**Theorem 4.2.** *For any $\epsilon > 0$, if the support of the distribution $\mathcal{P}_v^{(i)}$ is confined to a bounded space of $\mathbb{R}^d$, there exists a FNN $u(\cdot) : \mathbb{R}^d \to \mathbb{R}$ (and thus its gradient $\nabla u(\cdot) : \mathbb{R}^d \to \mathbb{R}^d$) with sufficiently large width and depth (depending on $\epsilon$) such that $\mathcal{W}_2^2(\mathcal{P}_v^{(i)}, \nabla u(\mathcal{G})) < \epsilon$ where $\nabla u(\mathcal{G})$ is the distribution generated through the mapping $\nabla u(\xi)$, $\xi \sim$ a $d$-dim non-degenerate Gaussian distribution.*

Another challenge is that the Wasserstein distance between $\psi_n(\boldsymbol{x}_{t-1}^i, t)$ and $\mathcal{P}_{\mathcal{N}_i^s}$ does not have a closed form. Thus, we utilize the empirical Wasserstein distance that can provably approximate the population one as in Peyré et al. [57]. For each forward pass, our CONPREDIFF will get $q$ sampled target pixel/feature points $\{\boldsymbol{x}_{(i,j)}^{tar} | 1 \leq j \leq q\}$ from $\mathcal{P}_{\mathcal{N}_i^s}$; Next, get $q$ samples from $\mathcal{N}(\mu_i, \Sigma_i)$, denoted by $\xi_1, \xi_2, ..., \xi_q$, and thus $\{\boldsymbol{x}_{(i,j)}^{pred} = \text{FNN}_n(\xi_j) | 1 \leq j \leq q\}$ are $q$ samples from the prediction $\psi_n(\boldsymbol{x}_{t-1}^i, t)$; Adopt the following empirical surrogated loss of $\mathcal{W}_2^2(\psi_n(\boldsymbol{x}_{t-1}^i, t), \mathcal{P}_{\mathcal{N}_i^s})$ in Equation (14):

$$
\min_\pi \sum_{j=1}^q \|\boldsymbol{x}_{(i,j)}^{tar} - \boldsymbol{x}_{(i,\pi(j))}^{pred}\|^2, \quad \text{s.t. } \pi \text{ is a bijective mapping:}[q] \to [q].
\tag{17}
$$

The loss function is based on solving a matching problem and needs the Hungarian algorithm with $O(q^3)$ complexity [33]. A more efficient surrogate loss may be needed, such as Chamfer loss based on greedy approximation [18] or Sinkhorn loss based on continuous relaxation [11], whose complexities are $O(q^2)$. In our study, as $q$ is set to a small constant, we use Equation (17) based on a Hungarian matching and do not introduce much computational overhead. The computational efficiency of design is empirically demonstrated in Sec. 5.3.

## 4.3 Discrete and Continuous CONPREDIFF

In training process, given previously-estimated $\boldsymbol{x}_t$, our CONPREDIFF simultaneously predict both $\boldsymbol{x}_{t-1}$ and the neighborhood distribution $\mathcal{P}_{\mathcal{N}_i^s}$ around each pixel/feature. Because $\boldsymbol{x}_{t-1}^i$ can be pixel, feature or discrete token of input image, we can generalize the CONPREDIFF to existing discrete and continuous backbones to form discrete and continuous CONPREDIFF. More concretely, we can substitute the point denoising part in Equation (14) alternatively with the discrete diffusion term $\mathcal{L}_{t-1}^{dis}$ (Equation (3)) or the continuous (Equation (6)) diffusion term $\mathcal{L}_{t-1}^{con}$ for generalization:

$$
\begin{aligned}
\mathcal{L}_{\text{CONPREDIFF}}^{dis} &= \mathcal{L}_{t-1}^{dis} + \lambda_t \cdot \sum_{i=1}^{x \times y} \mathcal{W}_2^2(\psi_n(\boldsymbol{x}_{t-1}^i, t), \mathcal{P}_{\mathcal{N}_i^s}), \\
\mathcal{L}_{\text{CONPREDIFF}}^{con} &= \mathcal{L}_{t-1}^{con} + \lambda_t \cdot \sum_{i=1}^{x \times y} \mathcal{W}_2^2(\psi_n(\boldsymbol{x}_{t-1}^i, t), \mathcal{P}_{\mathcal{N}_i^s}),
\end{aligned}
\tag{18}
$$

where $\lambda_t \in [0, 1]$ is a time-dependent weight parameter. Note that our CONPREDIFF only performs context prediction in training for optimizing the point denoising network $\psi_p$, and thus does not

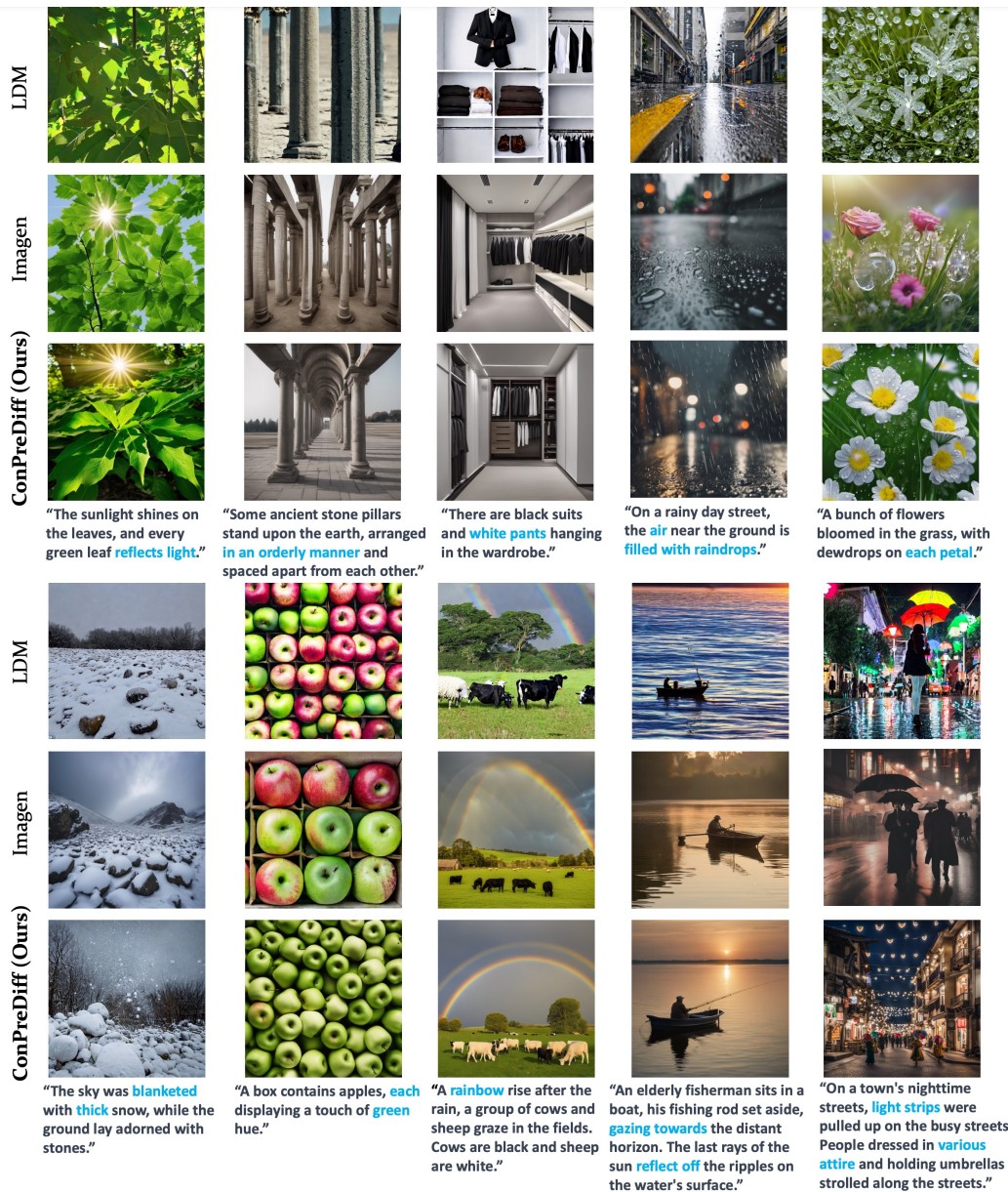

Figure 2: Synthesis examples demonstrating text-to-image capabilities of for various text prompts with LDM, Imagen, and ConPreDiff (Ours). Our model can better express local contexts and semantics of the texts marked in blue.

introduce extra parameters to the inference stage, which is computationally efficient. Equipped with our proposed context prediction term, existing diffusion models consistently gain performance promotion. Next, we use extensive experimental results to prove the effectiveness.

## 5 Experiments

### 5.1 Experimental Setup

**Datasets and Metrics** Regarding unconditional image generation, we choose four popular datasets for evaluation: CelebA-HQ [34], FFHQ [35], LSUN-Church-outdoor [102], and LSUN-bedrooms [102]. We evaluate the sample quality and their coverage of the data manifold using FID [26] and Precision-and-Recall [43]. For text-to-image generation, we train the model with LAION [73, 74]

Table 1: Quantitative evaluation of FID on MS-COCO for 256 × 256 image resolution.

| Approach | Model Type | FID-30K | Zero-shot FID-30K |
|----------|-----------|---------|-------------------|
| AttnGAN [96] | GAN | 35.49 | - |
| DM-GAN [113] | GAN | 32.64 | - |
| DF-GAN [86] | GAN | 21.42 | - |
| DM-GAN + CL [100] | GAN | 20.79 | - |
| XMC-GAN [107] | GAN | 9.33 | - |
| LAFITE [112] | GAN | 8.12 | - |
| Make-A-Scene [22] | Autoregressive | 7.55 | - |
| DALL-E [61] | Autoregressive | - | 17.89 |
| LAFITE [112] | GAN | - | 26.94 |
| LDM [65] | Continuous Diffusion | - | 12.63 |
| GLIDE [54] | Continuous Diffusion | - | 12.24 |
| DALL-E 2 [62] | Continuous Diffusion | - | 10.39 |
| Improved VQ-Diffusion [85] | Discrete Diffusion | - | 8.44 |
| Simple Diffusion [31] | Continuous Diffusion | - | 8.32 |
| Imagen [69] | Continuous Diffusion | - | 7.27 |
| Parti [104] | Autoregressive | - | 7.23 |
| Muse [7] | Non-Autoregressive | - | 7.88 |
| eDiff-I [3] | Continuous Diffusion | - | 6.95 |
| **CONPREDIFF**$_{dis}$ | Discrete Diffusion | - | **6.67** |
| **CONPREDIFF**$_{con}$ | Continuous Diffusion | - | **6.21** |

and some internal datasets, and conduct evaluations on MS-COCO dataset with zero-shot FID and CLIP score [25, 59], which aim to assess the generation quality and resulting image-text alignment. For image inpainting, we choose CelebA-HQ [34] and ImageNet [12] for evaluations, and evaluate all 100 test images of the test datasets for the following masks: Wide, Narrow, Every Second Line, Half Image, Expand, and Super-Resolve. We report the commonly reported perceptual metric LPIPS [109], which is a learned distance metric based on the deep feature space.

**Baselines**  To demonstrate the effectiveness of CONPREDIFF, we compare with the latest diffusion and non-diffusion models. Specifically, for unconditional image generation, we choose ImageBART[16], U-Net GAN (+aug) [72], UDM [39], StyleGAN [36], ProjectedGAN [71], DDPM [28] and ADM [14] for comparisons. As for text-to-image generation, we choose DM-GAN [113], DF-GAN [86], DM-GAN + CL [100], XMC-GAN [107] LAFITE [112], Make-A-Scene [22], DALL-E [61], LDM [65], GLIDE [54], DALL-E 2 [62], Improved VQ-Diffusion [85], Imagen-3.4B [69], Parti [104], Muse [7], and eDiff-I [3] for comparisons. For image inpainting, we choose autoregressive methods( DSI [56] and ICT [91]), the GAN methods (DeepFillv2 [103], AOT [105], and LaMa [84]) and diffusion based model (RePaint [49]). All the reported results are collected from their published papers or reproduced by open source codes.

**Implementation Details**  For text-to-image generation, similar to Imagen [68], our continuous diffusion model CONPREDIFF$_{con}$ consists of a base text-to-image diffusion model (64×64) [53], two super-resolution diffusion models [29] to upsample the image, first 64×64 → 256×256, and then 256×256 → 1024×1024. The model is conditioned on both T5 [60] and CLIP [59] text embeddings. The T5 encoder is pre-trained on a C4 text-only corpus and the CLIP text encoder is trained on an image-text corpus with an image-text contrastive objective. We use the standard Adam optimizer with a learning rate of 0.0001, weight decay of 0.01, and a batch size of 1024 to optimize the base model and two super-resolution models on NVIDIA A100 GPUs, respectively, equipped with multi-scale training technique (6 image scales). We generalize our context prediction to discrete diffusion models [24, 85] to form our CONPREDIFF$_{dis}$. For image inpainting, we adopt a same pipeline as RePaint [49], and retrain its diffusion backbone with our context prediction loss. We use T = 250 time steps, and applied r = 10 times resampling with jumpy size j = 10. For unconditional generation tasks, we use the same denoising architecture like LDM [65] for fair comparison. The max channels are 224, and we use T=2000 time steps, linear noise schedule and an initial learning rate of 0.000096.

Table 2: Quantitative evaluation of image inpainting on CelebA-HQ and ImageNet.

| **CelebA-HQ** Method | Wide LPIPS ↓ | Narrow LPIPS ↓ | Super-Resolve 2× LPIPS ↓ | Altern. Lines LPIPS ↓ | Half LPIPS ↓ | Expand LPIPS ↓ |
|---|---|---|---|---|---|---|
| AOT [105] | 0.104 | 0.047 | 0.714 | 0.667 | 0.287 | 0.604 |
| DSI [56] | 0.067 | 0.038 | 0.128 | 0.049 | 0.211 | 0.487 |
| ICT [91] | 0.063 | 0.036 | 0.483 | 0.353 | 0.166 | 0.432 |
| DeepFillv2 [103] | 0.066 | 0.049 | 0.119 | 0.049 | 0.209 | 0.467 |
| LaMa [84] | 0.045 | 0.028 | 0.177 | 0.083 | **0.138** | 0.342 |
| RePaint [49] | 0.059 | 0.028 | 0.029 | **0.009** | 0.165 | 0.435 |
| **CONPREDIFF** | **0.042** | **0.022** | **0.023** | 0.022 | **0.139** | **0.297** |

| **ImageNet** Method | Wide LPIPS ↓ | Narrow LPIPS ↓ | Super-Resolve 2× LPIPS ↓ | Altern. Lines LPIPS ↓ | Half LPIPS ↓ | Expand LPIPS ↓ |
|---|---|---|---|---|---|---|
| DSI [56] | 0.117 | 0.072 | 0.153 | 0.069 | 0.283 | 0.583 |
| ICT [91] | 0.107 | 0.073 | 0.708 | 0.620 | 0.255 | 0.544 |
| LaMa [84] | 0.105 | 0.061 | 0.272 | 0.121 | **0.254** | 0.534 |
| RePaint [49] | 0.134 | 0.064 | 0.183 | **0.089** | 0.304 | 0.629 |
| **CONPREDIFF** | **0.098** | **0.057** | **0.129** | 0.107 | 0.285 | **0.506** |

Our context prediction head contains two non-linear blocks (*e.g.*, Conv-BN-ReLU, resnet block or transformer block), and its choice can be flexible according to specific task. The prediction head does not incur significant training costs, and can be removed in inference stage without introducing extra testing costs. We set the neighborhood stride to 3 for all experiments, and carefully choose the specific layer for adding context prediction head near the end of denoising networks.

## 5.2 Main Results

**Text-to-Image Synthesis** We conduct text-to-image generation on MS-COCO dataset, and quantitative comparison results are listed in Tab. 1. We observe that both discrete and continuous CONPREDIFF substantially surpasses previous diffusion and non-diffusion models in terms of FID score, demonstrating the new state-of-the-art performance. Notably, our discrete and continuous CONPREDIFF achieves **an FID score of 6.67 and 6.21 which are better than the score of 8.44 and 7.27 achieved by previous SOTA discrete and continuous diffusion models**. We visualize text-to-image generation results in Figure 2, and find that our CONPREDIFF can synthesize images that are semantically better consistent with text prompts. It demonstrates our CONPREDIFF can make promising cross-modal semantic understanding through preserving visual context information generating process, and effectively associating with contextual text information. Moreover, we observe that CONPREDIFF can synthesize complex objects and scenes consistent with text prompts as demonstrated by Figure 7 in Appendix A.4, proving the effectiveness of our designed neighborhood context prediction. Human evaluations are provided in Appendix A.5.

**Image Inpainting** Our CONPREDIFF naturally fits image inpainting task because we directly predict the neighborhood context of each pixel/feature in diffusion generation. We compare our CONPREDIFF against state-of-the-art on standard mask distributions, commonly employed for benchmarking. As in Tab. 2, our CONPREDIFF outperforms previous SOTA method for most kinds of masks. We also put some qualitative results in Figure 3, and observe that CONPREDIFF produces a semantically meaningful filling, demonstrating the effectiveness of our context prediction.

**Unconditional Image Synthesis** We list the quantitative results about unconditional image generation in Tab. 3 of Appendix A.3. We observe that our CONPREDIFF significantly improves upon the state-of-the-art in FID and Precision-and-Recall scores on FFHQ and LSUN-Bedrooms datasets. The CONPREDIFF obtains high perceptual quality superior to prior GANs and diffusion models, while maintaining a higher coverage of the data distribution as measured by recall.

## 5.3 The Impact and Efficiency of Context Prediction

In Sec. 4.2, we tackle the complexity problem by transforming the decoding target from entire neighborhood features to neighborhood distribution. Here we investigate both impact and efficiency of the proposed neighborhood context prediction. For fast experiment, we conduct ablation study with the diffusion backbone of LDM [65]. As illustrated in Figure 4, the FID score of CONPREDIFF

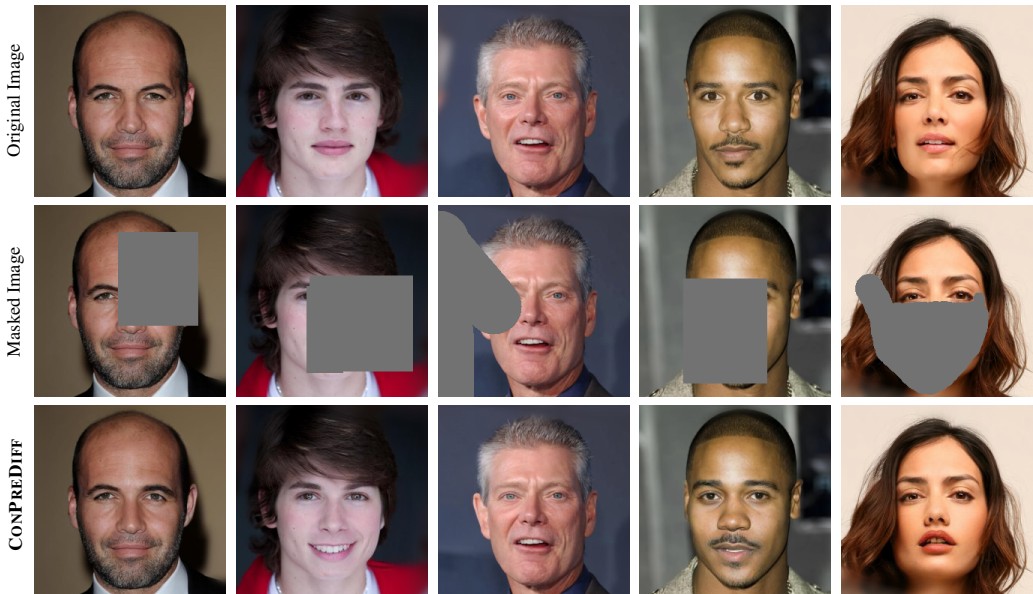

Figure 3: Inpainting examples generated by our CONPREDIFF.

is better with the neighbors of more strides and 1-stride neighbors contribute the most performance gain, revealing that preserving local context benefits the generation quality. Besides, we observe that increasing neighbor strides significantly increases the training cost when using feature decoding, while it has little impact on distribution decoding with comparable FID score. To demonstrate the generalization ability, we equip previous diffusion models with our context prediction head. From the results in Figure 5, we find that our context prediction can consistently and significantly improve the FID scores of these diffusion models, sufficiently demonstrating the effectiveness and extensibility of our method. We further conduct ablation study on the trade-off between CLIP and FID scores across a range of guidance weights in Appendix A.2, the results exhibit our superior generation ability.

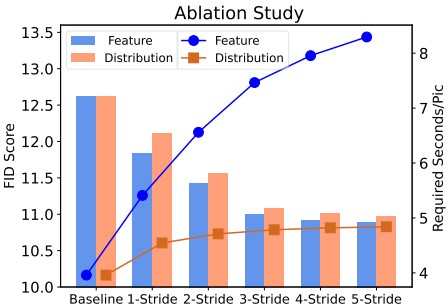

Figure 4: Bar denotes FID and line denotes time cost.

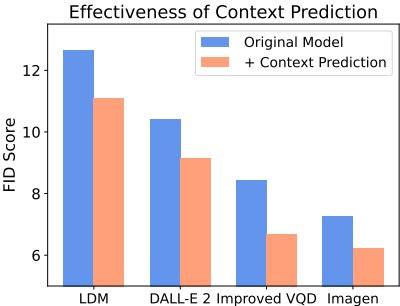

Figure 5: Equip diffusion models with our context prediction.

## 6  Conclusion

In this paper, we for the first time propose CONPREDIFF to improve diffusion-based image synthesis with context prediction. We explicitly force each point to predict its neighborhood context with an efficient context decoder near the end of diffusion denoising blocks, and remove the decoder for inference. CONPREDIFF can generalize to arbitrary discrete and continuous diffusion backbones and consistently improve them without extra parameters. We achieve new SOTA results on unconditional image generation, text-to-image generation and image inpainting tasks.

## Acknowledgement

This work was supported by the National Natural Science Foundation of China (No.61832001 and U22B2037).

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

# A  Appendix

## A.1  Limitations and Broader Impact

**Limitations**   While our ConPreDiff boosts performance of both discrete and continuous diffusion models without introducing additional parameters in model inference, our models still have more trainable parameters than other types of generative models, e.g GANs. Furthermore, we note the long sampling times of both and compared to single step generative approaches like GANs or VAEs. However, this drawback is inherited from the underlying model class and is not a property of our context prediction approach. Neighborhood context decoding is fast and incurs negligible computational overhead in training stage. For future work, we will try to find more intrinsic information to preserve for improving existing point-wise denoising diffusion models, and extend to more challenging tasks like text-to-3D and text-to-video generation.

**Broader Impact**   Recent generative image models enable creative applications and autonomous media creation, but can also be viewed as a dual-use technology with negative implications. In this paper, we use human face datasets only for evaluating the image inpainting performance of our method, and our method is not intended to create content that is used to mislead or deceive. However, like other related image generation methods, it could still potentially be misused in the realm of human impersonation. A notorious example are so-called "deep fakes" that have been used, for example, to create pornographic "undressing" applications. We strongly disapprove of any actions aimed at producing deceptive or harmful content featuring real individuals. Besides, generative methods have the capacity to be harnessed for other malicious intentions, including harassment and misinformation spread [20], and give rise to significant concerns pertaining to societal and cultural exclusion as well as biases [83, 82]. These considerations guide our decision not to release the source code or a public demo at this point in time.

Furthermore, the immediate availability of mass-produced high-quality images can be used to spread misinformation and spam, which in turn can be used for targeted manipulation in social media. Datasets are crucial for deep learning as they are the main input of information [101, 92, 93, 97]. Large-scale data requirements of text-to-image models have led researchers to rely heavily on large, mostly uncurated, web-scraped datasets. While this approach has enabled rapid algorithmic advances recently, datasets of this nature have been critiqued and contested along various ethical dimensions. One should consider the ability to curate the database to exclude (or explicitly contain) potential harmful source images. Creating a public API could offer a cheaper way to offer a safe model than retraining a model on a filtered subset of the training data or doing difficult prompt engineering. Conversely, including only harmful content is an easy way to build a toxic model.

## A.2  Guidance Scale *vs*. FID

To further demonstrate the effectiveness of our proposed context prediction, we quantitatively conduct evaluations about the trade-off between MS-COCO zero-shot FID [26] and CLIP scores. The results in Figure 6 indicate that the guidance hurts the diversity of GLIDE much more than DALL-E 2 and CONPREDIFF. The phenomenon reveals that the proposed CONPREDIFF can overall improve the generation quality of diffusion models.

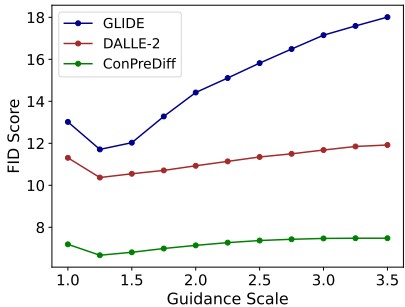

Figure 6: Trade-off between guidance scale and FID.

## A.3   More Quantitative Results

We list the unconditional generation results on FFHQ, CelebA-HQ, LSUN-Churches, and LSUN-Bedrooms in Tab. 3. We find CONPREDIFF consistently outperforms previous methods, demonstrating the effectiveness of the CONPREDIFF.

Table 3: Evaluation results for unconditional image synthesis.

| FFHQ 256 × 256 | | | |
|---|---|---|---|
| **Method** | FID ↓ | Prec. ↑ | Recall ↑ |
| ImageBART[16] | 9.57 | - | - |
| U-Net GAN (+aug) [72] | 7.6 | - | - |
| UDM [39] | 5.54 | - | - |
| StyleGAN [36] | 4.16 | 0.71 | 0.46 |
| ProjectedGAN [71] | 3.08 | 0.65 | 0.46 |
| LDM [65] | 4.98 | 0.73 | 0.50 |
| **CONPREDIFF** | **2.24** | **0.81** | **0.61** |
| LSUN-Bedrooms 256 × 256 | | | |
| **Method** | FID ↓ | Prec. ↑ | Recall ↑ |
| ImageBART [16] | 5.51 | - | - |
| DDPM [28] | 4.9 | - | - |
| UDM [39] | 4.57 | - | - |
| StyleGAN [36] | 2.35 | 0.59 | 0.48 |
| ADM [14] | 1.90 | 0.66 | 0.51 |
| ProjectedGAN [71] | 1.52 | 0.61 | 0.34 |
| LDM-4 [65] | 2.95 | 0.66 | 0.48 |
| **CONPREDIFF** | **1.12** | **0.73** | **0.59** |
| CelebA-HQ 256 × 256 | | | |
| **Method** | FID ↓ | Prec. ↑ | Recall ↑ |
| DC-VAE [55] | 15.8 | - | - |
| VQGAN+T. [17] (k=400) | 10.2 | - | - |
| PGGAN [43] | 8.0 | - | - |
| LSGM [87] | 7.22 | - | - |
| UDM [39] | 7.16 | - | - |
| LDM [65] | 5.11 | 0.72 | 0.49 |
| **CONPREDIFF** | **3.22** | **0.83** | **0.57** |
| LSUN-Churches 256 × 256 | | | |
| **Method** | FID ↓ | Prec. ↑ | Recall ↑ |
| DDPM [28] | 7.89 | - | - |
| ImageBART [16] | 7.32 | - | - |
| PGGAN [43] | 6.42 | - | - |
| StyleGAN [36] | 4.21 | - | - |
| StyleGAN2 [37] | 3.86 | - | - |
| ProjectedGAN [71] | **1.59** | 0.61 | 0.44 |
| LDM [65] | 4.02 | 0.64 | 0.52 |
| **CONPREDIFF** | **1.78** | **0.74** | **0.61** |

## A.4   More Synthesis Results

We visualize more text-to-image synthesis results on MS-COCO dataset in Figure 7. We observe that compared with previous powerful LDM and DALL-E 2, our CONPREDIFF generates more natural and smooth images that preserve local continuity.

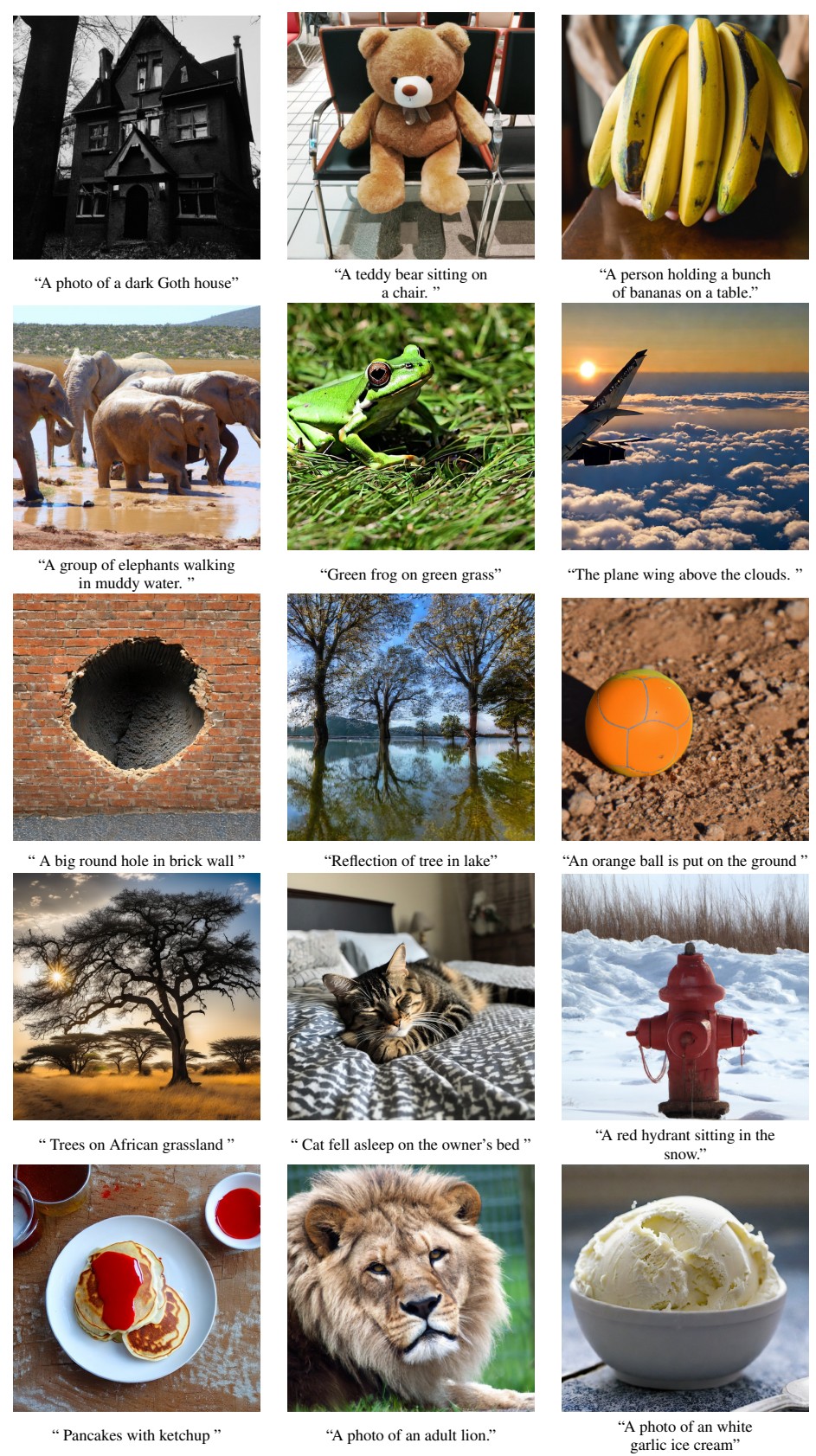

Figure 7: Synthesis examples demonstrating text-to-image capabilities of for various text prompts.

## A.5 Human Evaluations

As demonstrated in qualitative results, our CONPREDIFF is able to synthesize realistic diverse, context-coherent images. However, using FID to estimate the sample quality is not always consistent with human judgment. Therefore, we follow the protocol of previous works [104, 68, 62], and conduct systematic human evaluations to better assess the generation capacities of our CONPREDIFF from the aspects of image photorealism and image-text alignment. We conduct side-by-side human evaluations, in which well-trained users are presented with two generated images for the same prompt and need to choose which image is of higher quality and more realistic (image photorealism) and which image better matches the input prompt (image-text alignment). For evaluating the coherence of local context, we propose a new evaluation protocol, in which users are presented with 1000 pairs of images and must choose which image better preserves local pixel/semantic continuity. The evaluation results are in Tab. 4, CONPREDIFF performs better in pairwise comparisons against both Improved VQ-Diffusion and Imagen. We find that CONPREDIFF is preferred in terms of all three evaluations, and CONPREDIFF is strongly preferred regarding context coherence, demonstrating that preserving local neighborhood context advances sample quality and semantic alignment.

Table 4: Human evaluation comparing CONPREDIFF to Improved VQ-Diffusion and Imagen.

|  | Improved VQ-Diffusion | Imagen |
|---|---|---|
| Image Photorealism | 72% | 65% |
| Image-Text Alignment | 68% | 63% |
| Context Coherence | 84% | 78% |

