# OpenReview forum: "Improving Diffusion-Based Image Synthesis with Context Prediction"
_NeurIPS.cc/2023/Conference — NeurIPS 2023 poster_

### Official Review · Reviewer_2mXr · 2023-06-14

**Soundness:** 3 good
**Presentation:** 2 fair
**Contribution:** 2 fair
**Rating:** 6
**Confidence:** 5

**Summary:**

This paper proposes to improve diffusion-based image synthesis by explicitly reinforcing each point to predict its neighborhood context during training, without extra cost at inference. To reduce computation/time complexity of context decoding the authors propose efficient large context decoding adopting Wasserstein distance to characterize the distribution reconstruction loss. The method is applicable to both discrete and continuous diffusion backbones and achieves new SOTA text-to-image generation on MS-COCO with FID 6.21

**Strengths:**

1. The paper is well-written and easy to follow.
2. The method of explicitly reinforcing each point to predict its neighborhood context for diffusion models is well-motivated with effective designs to reduce substantial computation complexity for large-context neighborhood reconstruction.
3. The method is proven effective in boosting FID scores on MS-COCO text-to-image synthesis for both continuous (eDiff-I) and discrete diffusion (VQ-Diffusion) backbones.

**Weaknesses:**

1. The main results are on text-to-image synthesis and image inpainting. It would be good to add unconditional generation results.
2. The method emphasizes on diffusion with better neighbouring context, leading to generations "semantically better consistent with the text prompts"(L233), "prommising cross-modal semantic understanding" (L234), "can synthesize more complex objects and scenes" (L236-237). I don't think the claims are well-justified: e.g. Fig. 2 and Fig. 3 only presents results of the proposed method without any comparison to warrant the aforementioned conclusions. More analysis and evidence of "better semantics" are required other than the overall FID score.
3. Some notations are misleading, e.g. L 114-116, for h_i (h_{t-1}), the subscript is used to indicate both spatial and time; x_t is not defined in the main text.

**Questions:**

1. L260-262: regarding to Fig. 5 I'm not convinced by the observation and conclusion, could the authors make it more clear ?
2. I'm curious about the comparision with Dalle-2 and Imagen in Fig. 7 as the models are not open-sourced.

**Limitations:**

No.

---

> ### Author Rebuttal · Authors · 2023-08-06
>
> *We thank Reviewer 2mXr for the positive review and valuable feedback. We are glad that the reviewer found that the paper is well-written and easy to follow, the method is well-motivated with effective designs, and the method is proven effective in boosting FID scores with both continuous and discrete diffusion backbones. Please see below for our responses to your comments.*
>
> **Q1: The main results are on text-to-image synthesis and image inpainting. It would be good to add unconditional generation results. More evidence of "better semantics" are required other than the overall FID score.**
>
> A1: We have listed unconditional generation results on four datasets in the appendix, and consistently outperform previous methods. More qualitative comparison results for demonstrating  "better semantics" of our method can be found in **global response pdf**. From the visual comparison, we conclude that our ConPreDiff can sufficiently capture the semantics in the text prompt, and can better express them in the generated images compared to powerful LDM and Imagen.
>
> **Q2: Some notations are misleading, e.g. L 114-116, for $h_i (h_{t-1})$, the subscript is used to indicate both spatial and time; x_t is not defined in the main text.**
>
> A2: $h_{t-1}$ and $x_t$ are used to denote the predicted point and previously-predicted image in the diffusion process, respectively. We will make the notations more clear, thanks for your constructive suggestions.
>
> **Q3: L260-262: regarding to Fig. 5 I'm not convinced by the observation and conclusion, could the authors make it more clear?**
>
> A3: As demonstrated in Fig.5, the neighborhood decoding error, the point-wise reconstruction error, and FID score consistently decline in the training process. The phenomenon means that our "context prediction" term and standard "point-wise reconstruction" term are cooperative, they simultaneously make better FID score. We theoretically prove this cooperation (from the perspective of maximizing ELBO/likelihood) as the following:
>
> At time t:
>
> Loss of DDPM ( after reparamterization and scaling): $||x_0-\hat{x}_0(x_t,t)||_2^2$
>
> Loss of ConPreDiff:  $\sum\_{i = 1}^{x*y} [\mathcal{M}\_p(x\_0^{i} ,\hat{x}\_0^{i})+ \mathcal{M}\_n(\mathcal{H}\_{\mathcal{N}\_i},\hat{\mathcal{H}}\_{\mathcal{N}\_i})]$
>
> We let $\mathcal{M}_p,\mathcal{M}_n$ be square loss,
>   $\mathcal{M}_n(\mathcal{H}\_{\mathcal{N}\_i},\hat{\mathcal{H}}\_{\mathcal{N}\_i})=\sum\_{j\in \mathcal{N}\_i}(x\_0^{i,j}-\hat{x}\_0^{i,j})^2$,
>
>   where $x_0^{i,j}$ is the j-th neighbor in the context of $x_0^i$ and $\hat{x}_0^{i,j}$ is the prediction of $x_0^{i,j}$ from a denoising neural network. Following the notation in the main paper, we have
>
>   $\hat{x}_0^{i,j} = \psi_n(\psi_p(x_t,t)(i))(j)$,
>
>   where $\psi_p(x_t,t)(i)$ is the prediction of $x_0^i$ and $\psi_n$ is the neighborhood decoder. Compactly, we can write the denoising network as:
>
>   $$\Psi(x_t,t)(i,j) =\left\\{ \begin{array}{l}\psi_n(\psi_p(x_t,t)(i))(j), & j \in \mathcal{N}\_i, \\\\ \psi_p(x_t,t)(i), & j=i \end{array} \right.$$
>
>   We can show that the DDPM loss is upper bounded by ConPreDiff loss, by reparameterizing $\hat{x}_0(x_t,t)$.
>   Specifically, for each unit i in the feature map, we predict the unit i and its neighbors using equation (1), and then we use the mean of  predicted value in the neighborhood as the final prediction:
>
>   $$\hat{x}\_0(x_t,t)(i) = 1/(|\mathcal{N}\_i|+1)*\sum\_{j \in \mathcal{N}\_{i}\ \cup\ \\{i\\}}\Psi(x\_t,t)(i,j)$$
>
>   Now we can show the connection between DDPM loss with the mean prediction and ConPreDiff loss:
>
> $$\begin{array}{rl}||x_0-\hat{x}\_0(x_t,t)||_2^2 & = \sum_i (x_0^i-\hat{x}\_0(x_t,t)(i))^2,\\\\ &=\sum_i (x_0^i-\sum\_{j \in \mathcal{N}_i\cup\{i\}}\Psi(x_t,t)(i,j)/(|\mathcal{N}_i|+1))^2,\\\\ &=\sum_i(\sum\_{j \in \mathcal{N}_i\cup\{i\}}(\Psi(x_t,t)(i,j)-x_0^i))^2/(|\mathcal{N}_i|+1)^2, \\\\ (Cauchy\ inequality) & \leq \sum_i \sum\_{j \in \mathcal{N}\_i\cup\{i\}} (\Psi(x_t,t)(i,j)-x_0^i)^2/(|\mathcal{N}_i|+1), \\\\ &=1/(|\mathcal{N}\_i|+1)\sum_i [(x_0^i- \psi_p(x_t,t)(i))^2+\sum\_{j \in \mathcal{N}\_i} (x_0^{i,j}-\hat{x}\_0^{i,j})^2] \end{array}$$
>
>
>   in the last equality, we assume that the feature is padded so that each unit i has the same number of neighbors $|\mathcal{N}|$.
>   As a result, the ConPreDiff loss is an upper bound of the negative log likelihood. Thus both terms are cooperative for optimizing the generation quality of the model.
>
> **Q4: The comparison with Dalle-2 and Imagen.**
>
> A4: For the results of Dalle-2 and Imagen, we directly adopt published results from the papers for fair comparison. We adopt the codes reproduced by open source community, and further modify them to implement our ConPreDiff. And we are committed to open sourcing all the codes and trained models for all datasets and all tasks upon the acceptance.

---

> > ### Comment · Reviewer_2mXr · 2023-08-17
> >
> > Thanks for the response, I've raised my rating accordingly.

---

### Official Review · Reviewer_ydZz · 2023-06-28

**Soundness:** 3 good
**Presentation:** 3 good
**Contribution:** 3 good
**Rating:** 6
**Confidence:** 3

**Summary:**

This paper proposes to improve diffusion based image generative training objectives by adding context prediction loss. The motivation of predicting context comes from other non-diffusion based models like semantic segmentation and representation learning. To mitigate the complexity of predicting large per pixel neighbourhood context, the author further models the context as a probability distribution using Wasserstein distance. Experiments show the proposed model achieves new SoTA generation on MSCOCO for both discrete and continuous diffusion models.


**Strengths:**

The introduction is well-written and the motivation of predicting context in the diffusion based model is easy to follow.

The presentation of the method including the loss derivation and the training pipeline is easy to understand.

The authors conduct intensive experiments showing the proposed context prediction loss can be used on various DM models, achieving SoTA performance on MSCOCO FID and inpainting tasks.


**Weaknesses:**

The paper lacks training and implementation details. For example, the text-to-image experiment uses T5 encoder as the text encoder, but did not mention architecture details and training details.

One big motivation to model context as a probability distribution is to improve the training efficiency. As shown in Figure 6, feature matching has lower throughput compared to distribution matching, but it has better FID. I think an important baseline is missing – sampling based feature matching, i.e. use the same number of random samples as proposed in the distribution matching, 9 instead of the full neighbourhoods features for context prediction.


**Questions:**

In Table 1, it is not clear which diffusion model is used with the proposed method for both discrete and continuous diffusion models.

For qualitative comparison in Figure 2 and 8, there is no head to head comparison with other baselines, and it is hard to appreciate the improvement.


**Limitations:**

There is no limitation/future work discussion in the paper

And there is also no training / implementation details in the paper, which cause concerns on reproducibility.

---

> ### Author Rebuttal · Authors · 2023-08-06
>
> *We thank Reviewer ydZz for the positive review and valuable feedback. We are glad that the reviewer found that the introduction is well-written, the motivation is easy to follow, the presentation of the method including the loss derivation and the training pipeline is easy to understand, and the conducted experiments are extensive. Please see below for our responses to your comments.*
>
> **Q1: The paper lacks training and implementation details. For example, the text-to-image experiment uses T5 encoder as the text encoder, but did not mention architecture details and training details.**
>
> A1: We provide some important implementation details in section 4.1. Because our ConPreDiff can generalize to both continuous and discrete diffusion models, our training and implementation details mainly follow the diffusion models that we generalize to for fair comparisons. Here we provide more details for clarity:
>
> For text-to-image tasks, like Imagen, an Efficient U-Net architecture with 22 resnet blocks at total is utilized. A frozen T5-xl encoder is utilized as our text encoder. The text encoder is a 12 layers Transformer with 32-head multi-head attention and is pre-trained on a C4 text-only corpus with our context prediction based denoising objective. We use the standard Adam optimizer with 1e-4 learning rate.  We use the same cosine noise schedule as the Imporved DDPM [1]. For inpainting tasks, we adopt a same pipeline as RePaint, and a smaller U-Net architecture with 12 resnet blocks at total is utilized. The image size is 256x256. We use T = 250 time steps, and applied r = 10 times resampling with jumpy size j = 10.  For unconditional generation tasks, like LDM, we use a 8 layers U-Net. The max channels are 224. We use T=2000 time steps and linear noise schedule. The initial learning rate is 9.6e-5.
>
> **We are committed to open sourcing all the codes and trained models for all datasets and all tasks upon the acceptance.**
>
> [1] Nichol A Q, Dhariwal P. Improved denoising diffusion probabilistic models[C]//International Conference on Machine Learning. PMLR, 2021: 8162-8171.
>
> **Q2: Adding another baseline – sampling based feature matching, i.e. use the same number of random samples as proposed in the distribution matching, instead of the full neighborhood features for context prediction.**
>
> A2: Following your suggestions, we conduct fast experiments with sampling based feature matching, on the CelebA-HQ dataset. Compared with our approach, it predicts neighbors in a faster way, but it performs much worse than our "context prediction" which is based on distribution decoding, because our ConPreDiff adopts a surrogated loss of the entire neighborhoods matching.  And our approach can achieve a better trade-off between the FID score and training cost.
> |CelebA-HQ |LDM| LDM + Context Prediction| LDM + Sampling-based Feature Matching|
> | :-----| :----: | :----: |:----: |
> |Training time （sec/step） | 4.02 | 4.79| 4.22 |
> |FID score| 5.11 | **3.22**| 4.35 |
>
>
> **Q3: In Table 1, it is not clear which diffusion model is used with the proposed method for both discrete and continuous diffusion models.**
>
> A3: In the implementation details of section 4.1, we introduce that we generalize our ConPreDiff to discrete diffusion models (Improved VQ-Diffusion) to form our discrete $\text{ConPreDiff}\_{dis}$ and to continuous diffusion models (DALL-E 2, Imagen) to form our continuous $\text{ConPreDiff}\_{con}$.
>
> **Q4: More qualitative comparisons to appreciate the improvement.**
>
> A4: More qualitative comparison results can be found in **global response pdf**. From the results, we can easily find that our ConPreDiff can better express the local contexts and consistent semantics in the generated images compared to other methods.
>
>
> **Q5: Limitation/future work.**
>
> A5: We will add these in final version.
>
> Limitation/future work:
>
> While our ConPreDiff boosts performance of both discrete and continuous diffusion models without introducing additional parameters in model inference, our models still have more trainable parameters than other types of generative models, e.g GANs. Futhermore, we note the long sampling times of both $\text{ConPreDiff}\_{dis}$ and $\text{ConPreDiff}\_{con}$ compared to single step generative approaches like GANs or VAEs. However, this drawback is inherited from the underlying model class and is not a property of our context prediction approach. Neighborhood context decoding is fast and incurs negligible computational overhead in training stage. For future work, we will try to find high-order and more intrinsic information to preserve for improving existing point-wise denoising diffusion models.

---

> > ### Comment · Reviewer_ydZz · 2023-08-15
> >
> > Thank the authors for the response. The response addressed my concerns and I would like to keep my original rating.

---

### Official Review · Reviewer_DpwQ · 2023-07-06

**Soundness:** 3 good
**Presentation:** 3 good
**Contribution:** 3 good
**Rating:** 5
**Confidence:** 3

**Summary:**

This paper presents ConPreDiff, a method introduced to improve the performance of diffusion models by preserving the neighborhood context of predicted pixels/features. They achieve this by predicting the neighborhood context during the diffusion generation process. To simplify the modeling complexity, they propose predicting distributions instead of directly reconstructing the neighborhood. The method's effectiveness is demonstrated through extensive experiments on unconditional image generation, text-to-image generation, and image inpainting.

**Strengths:**

* The idea is intuitive and easy to understand.
* The proposed method is general and can be easily applied to recent diffusion models.
* The performance of the proposed method is very impressive.

**Weaknesses:**

* Recent diffusion models use UNet backbone, which stacks many convolutional and self-attention layers. Thus, it has a large receptive field. Additionally, LDM also has a decoder, which also has a decent receptive field. Therefore, I am confused about the paper's main claim that the point-wise reconstruction neglects to fully preserve the local context.
* There are no visual comparisons of the proposed method and baselines. I am not sure if ConPreDiff can really be more local-context consistent compared to other methods.
* The authors need to discuss the additional training cost. Besides, they also need to provide the additional parameters they use for the context prediction.

**Questions:**

* L138: Why do you choose $\mathcal P_{\mathcal N_i^s}$ in this form? Is there any insight that motivates you doing this?
* L115: What is $\mathbf h_{t-1}$? I suppose it should be $\mathbf x_{t-1}$.
* L119: What is $\mathbf h_{t-1}$? I think it should be $\mathbf h_i$.
* L138: What is $h_u^{(0)}$?
* Figure 4: What is NDM?

**Limitations:**

The authors did not discuss their limitations and societal impact in the paper.

---

> ### Author Rebuttal · Authors · 2023-08-06
>
> *We thank Reviewer DpwQ for your valuable feedback. We are glad that the reviewer found that our idea is intuitive and easy to understand, the proposed method is general and can be easily applied to recent diffusion models, and the performance of the proposed method is very impressive. Please see below for our responses to your comments.*
>
> **Q1: Being confused about the paper's main claim that the point-wise reconstruction neglects to fully preserve the local context.**
>
> A1: Although existing UNet-based diffusion models has a decent receptive filed, the process of acquiring receptive field is progressive, where the each point's local context information of previous layers may partially reduce or be filtered by multiple non-linear and pooling functions. Thus we explicitly add "context prediction" term at the end of the denoising network to explicitly reconstruct local context for each point，and maximally preserve useful context information from the perspective of reconstructing neighborhood distribution.
>
> **Q2: More visual comparison.**
>
> A2: More visualization comparison results can be found in **global response pdf**, from which we can easily find that our ConPreDiff can better express the local contexts and consistent semantics of text prompts in the generated images compared to other methods.
>
> **Q3: The authors need to discuss the additional training cost. Besides, they also need to provide the additional parameters they use for the context prediction.**
>
> A3: We report the approximate time cost (seconds) of these models to train in a single step on the MS-COCO dataset in the table below. The additional context prediction head (two linear-BN-ReLU modules) added to the models accounts for **only a small portion of the parameters (around 0.13M)**, and the neighborhood distribution decoding of our method also does not incur significant additional training costs. Notably, our "context prediction" head can be removed in inference stage without introducing extra testing costs.
> |Training cost (secs/step) |LDM | DALL·E 2| Improved VQ-Diffusion| Imagen|
> | :-----| :----: | :----: |:----: |:----: |
> |Original Model  | 11.9 | 74.7 | 83.9 |198.3 |
> | + Context Prediction | 13.1 | 77.5 |87.6 |206.3 |
>
>
> **Q4: (1)** Why do you choose P_NS in this form? Is there any insight that motivates you doing this? **(2)** L115: What is $h_{t-1}$? I suppose it should be $x_{t-1}$. **(3)** L119: What is $h_{t-1}$? I think it should be $h_i$. **(4)** L138: What is $h_{u}^{(0)}$? Figure 4: What is NDM?
>
> A4: **(1)** In L138, $P_{N^s_i}$ is defined by uniform distribution, which means we sample neighbors from local context using equal probabilities. The main insight is after the distribution decoding, neighbors are sampled without spatial orders and ground truth neighbors in a local area are all important to local semantics. Thus we equally treat sampled neighbors for reconstruction.
>
> **(2)** $h_{t-1}$ denotes one point in $x_{t-1}$, which is equal to $h_i$. And we represent each point of $x_{t-1}$ in both time form ($h_{t-1}$) and spatial form ($h_{i}$) for better explanations.
>
> **(3)** "(0)" is the superscript of $h_u$, which denotes the center point of neighborhoods $N^s_{i}$ (i.e., $h_i$).
>
> **(4)** NDM is a typo, we previously name our model Neighborhood Diffusion Model (NDM).
>
>
> **Q5: The authors did not discuss their limitations and societal impact in the paper.**
>
> A5:  We will add these in final version
>
> Limitations:
>
> While our ConPreDiff boosts performance of both discrete and continuous diffusion models without introducing additional parameters in model inference,
> our models still have more trainable parameters than other types of generative models, e.g GANs. Futhermore, we note the long sampling times of both $\text{ConPreDiff}\_{dis}$ and $\text{ConPreDiff}\_{con}$ compared to single step generative approaches like GANs or VAEs. However, this drawback is inherited from the underlying model class and is not a property of our context prediction approach. Neighborhood context decoding is fast and incurs negligible computational overhead in training stage. For future work, we will try to find high-order and more intrinsic information to preserve for improving existing point-wise denoising diffusion models.
>
> Broader Impact:
>
> Recent generative image models enable creative applications and autonomous media creation, but can also be viewed as a dual-use technology with negative implications. In this paper, we use human face datasets only for evaluating the image inpainting performance of our method, and our method is not intended to create content that is used to mislead or deceive. However, like other related image generation techniques, it could still potentially be misused for impersonating humans.  A notorious example are so-called “deep fakes” that have been used, for example, to create pornographic “undressing” applications. We condemn any behavior to create misleading or harmful content of a real person. Furthermore, the immediate availability of mass-produced high-quality images can be used to spread misinformation and spam, which in turn can be used for targeted manipulation in social media. Datasets are crucial for deep learning as they are the main input of information.
> Large scale data requirements of text-to-image models have led researchers to rely heavily on large, mostly uncurated, web-scraped datasets. While this approach has enabled rapid algorithmic advances in recent years, datasets of this nature have been critiqued and contested along various ethical dimensions. Furthermore, one should consider the ability to curate the database to exclude (or explicitly contain) potential harmful source images. When creating a public API that approach could offer a cheaper way to offer a safe model than retraining a model on a filtered subset of the training data or doing difficult prompt engineering. Conversely, including only harmful content is an easy way to build a toxic model.

---

> > ### Comment · Reviewer_DpwQ · 2023-08-17
> > **Official Comment by Reviewer DpwQ**
> >
> > Thank authors for their response. My major concerns are well addressed, but I still get confused by some notations in the paper, such as the $h_{t-1}$ and $\delta_{h_u^{(0)}}$, as I mentioned in the review.
> > * How can $h_{t-1}$ represent a point in $x_{t-1}$? By the definition, it is $t-1$-th feature point of the feature map.
> > * Is $\delta$ dirac delta function? Why do you need to denote the center point of $N_i^s$ (i.e., $h_i$) as $h_{u}^{(0)}$?
> >
> > I suggest the authors carefully proofread their manuscript, especially the notations. I will raise my score by 1.

---

> > > ### Author Response · Authors · 2023-08-18
> > > **Response to Reviewer DpwQ**
> > >
> > > We sincerely thank Reviewer DpwQ for raising score. For $h_i (h_{t-1})$, the subscript is used to denote spatial and time information, respectively, because we want to illustrate the previous point-based diffusion process from different perspectives. For better illustration, we will use $x_{t-1}^i$ to denote the $i$-th feature point at time step $t-1$. $\delta$ denotes the dirac delta function, and the notation of center point can be optionally removed. Following your suggestion, we will carefully proofread our manuscript. Thanks for your kind response.

---

### Official Review · Reviewer_qzoS · 2023-07-07

**Soundness:** 4 excellent
**Presentation:** 4 excellent
**Contribution:** 4 excellent
**Rating:** 6
**Confidence:** 5

**Summary:**

This paper proposes an idea of context prediction to boost difussion-based image generation.
The core idea is that in each step of diffusion, after the denoised point is generated, neighborhood context prediction is performed.
In particular, to maintain the spatial orders of the neighborhood, a permutation invariant loss is used for optimization by replacing the context prediction with neighborhood distribution prediction. Performance improvement against standard diffusion models were presented. in experiments.


**Strengths:**

1. The idea is very interesting and sound.
From an image denoising point of view, neighborhood info is commonly used, so it's a natural extension of diffusion-denoising models.
2. Performance improvement showed in experiments are promising.

**Weaknesses:**

The proposed approach probably takes longer to train. Can you discuss from that perspective?

**Questions:**

1. By introducing context, is any sign of "blurriness" introduced?
2. What are the typical cases that do worse compared to standard diffusion?

**Limitations:**

Similar with text2image papers.

---

> ### Author Rebuttal · Authors · 2023-08-06
>
> *We thank Reviewer qzoS for the positive review and valuable feedback. We are glad that the reviewer found that the idea is very interesting and sound, and performance improvement showed in experiments is promising. Please see below for our responses to your comments.*
>
> **Q1: The proposed approach probably takes longer to train. Can you discuss from that perspective?**
>
> A1: Thanks for your interesting question, we here discuss this problem from two training scenarios:
>
> 1.Training ConPreDiff from scratch
>
> We introduce the "context prediction" term as in Eq.1. Actually the proposed "context prediction" term plays a additional role in maximizing ELBO and will accelerate the convergence (i.e. less training steps) of diffusion models. We here provide the formula derivation：
>
> At time t:
>
> Loss of DDPM ( after reparamterization and scaling): $||x_0-\hat{x}_0(x_t,t)||_2^2$
>
> Loss of ConPreDiff:  $\sum\_{i = 1}^{x*y} [\mathcal{M}\_p(x\_0^{i} ,\hat{x}\_0^{i})+ \mathcal{M}\_n(\mathcal{H}\_{\mathcal{N}\_i},\hat{\mathcal{H}}\_{\mathcal{N}\_i})]$
>
> We let $\mathcal{M}_p,\mathcal{M}_n$ be square loss,
>   $\mathcal{M}_n(\mathcal{H}\_{\mathcal{N}\_i},\hat{\mathcal{H}}\_{\mathcal{N}\_i})=\sum\_{j\in \mathcal{N}\_i}(x\_0^{i,j}-\hat{x}\_0^{i,j})^2$,
>
>   where $x_0^{i,j}$ is the j-th neighbor in the context of $x_0^i$ and $\hat{x}_0^{i,j}$ is the prediction of $x_0^{i,j}$ from a denoising neural network. Following the notation in the main paper, we have
>
>   $\hat{x}_0^{i,j} = \psi_n(\psi_p(x_t,t)(i))(j)$,
>
>   where $\psi_p(x_t,t)(i)$ is the prediction of $x_0^i$ and $\psi_n$ is the neighborhood decoder. Compactly, we can write the denoising network as:
>
>   $$\Psi(x_t,t)(i,j) =\left\\{ \begin{array}{l}\psi_n(\psi_p(x_t,t)(i))(j), & j \in \mathcal{N}\_i, \\\\ \psi_p(x_t,t)(i), & j=i \end{array} \right.$$
>
>   We can show that the DDPM loss is upper bounded by ConPreDiff loss, by reparameterizing $\hat{x}_0(x_t,t)$.
>   Specifically, for each unit i in the feature map, we predict the unit i and its neighbors using equation (1), and then we use the mean of  predicted value in the neighborhood as the final prediction:
>
>   $$\hat{x}\_0(x_t,t)(i) = 1/(|\mathcal{N}\_i|+1)*\sum\_{j \in \mathcal{N}\_{i}\ \cup\ \\{i\\}}\Psi(x\_t,t)(i,j)$$
>
>   Now we can show the connection between DDPM loss with the mean prediction and ConPreDiff loss:
>
> $$\begin{array}{rl}||x_0-\hat{x}\_0(x_t,t)||_2^2 & = \sum_i (x_0^i-\hat{x}\_0(x_t,t)(i))^2,\\\\ &=\sum_i (x_0^i-\sum\_{j \in \mathcal{N}_i\cup\{i\}}\Psi(x_t,t)(i,j)/(|\mathcal{N}_i|+1))^2,\\\\ &=\sum_i(\sum\_{j \in \mathcal{N}_i\cup\{i\}}(\Psi(x_t,t)(i,j)-x_0^i))^2/(|\mathcal{N}_i|+1)^2, \\\\ (Cauchy\ inequality) & \leq \sum_i \sum\_{j \in \mathcal{N}\_i\cup\{i\}} (\Psi(x_t,t)(i,j)-x_0^i)^2/(|\mathcal{N}_i|+1), \\\\ &=1/(|\mathcal{N}\_i|+1)\sum_i [(x_0^i- \psi_p(x_t,t)(i))^2+\sum\_{j \in \mathcal{N}\_i} (x_0^{i,j}-\hat{x}\_0^{i,j})^2] \end{array}$$
> As a result, the ConPreDiff loss is an upper bound of the negative log likelihood. Thus both terms are cooperative for convergence, and adding "context prediction" term would accelerate the convergence with less training steps.
>
> 2.Training ConPreDiff from pre-trained diffusion models
>
>  Initializing ConPreDiff with any pre-trained diffusion models will need additional fine-tuning time for adjusting base denoising network and training our context prediction head.
>
> **Q2: By introducing context, is any sign of "blurriness" introduced?**
>
> A2: There is no sign of "blurriness" by introducing context, which can be partially explained by above derivations. We also put more qualitative comparison results  **in global response pdf**. From the results, we can easily find that our ConPreDiff can better generate the images semantically consistent with text prompts compared to other methods, and preserve local context well without any sign of "blurriness".
>
> **Q3: What are the typical cases that do worse compared to standard diffusion?**
>
> A3: Currently, we do not find obvious cases or typical cases that do worse than standard diffusion. However, we will continue to improve our model, explore more potential applications, and address possible limitations for future work.

---

> > ### Comment · Reviewer_qzoS · 2023-08-20
> >
> > Thanks authors for the response. The response addressed my concerns. I suggest to accept this paper.

---

### Official Review · Reviewer_m5Qc · 2023-07-12

**Soundness:** 3 good
**Presentation:** 3 good
**Contribution:** 3 good
**Rating:** 6
**Confidence:** 4

**Summary:**

This paper is proposing context-aware Diffusion Models. They make the models learn the context information by setting up auxiliary networks to estimate the neighbor distributions from the estimated denoised sample from Diffusion Models. The benefit of this approach is that additional cost from the auxiliary networks are not applied during the sampling. Both quantitative and qualitative experiments are reported.

**Strengths:**

- Motivation is agreeable.
- Good writing.
- Reasonable method for motivation.
- Experiments are done well.

**Weaknesses:**

Specifics of the weaknesses of this paper are written below as questions and limitations, but I believe most of them can be resolved during the rebuttal. I will increase my rating if my concerns can be resolved.

**Questions:**


* The meaning of "FNN" is not defined.
* What is the meaning of “(0)” of the delta function in line 138?
* Why not let $FNN_{\mu}$ and $FNN_{\sigma}$ take target index as additional input and skip Eq. 7? Once the target index is specified, the matching algorithm may not be needed anymore.
* What is the relationship between $q$ in Eq. 7 and stride $s$ and $K$?
* It would be better if performance of directly estimating the neighbors is reported as well since it would be more accurate and simple setup to implement the motivation of this paper (even though it has the memory inefficiency, as mentioned in L132).
* Can it be described more specifically why KL or JSD cannot be applied (L155)? It is not straightforward to me why KL cannot be applied to this task.
* Are both terms in Eq. 3 used together for finetuning?
* What happens if Diffusion Models are trained from scratch with the proposed objectives?


**Limitations:**

- The most important part of the paper would be “context prediction” term in Eq . 3. It’s motivation is understandable, but it is not interpreted in terms of optimizing variational bound of negative log likelihood. Is it just “additional” term? or can it be interpreted as a term playing a certain role in maximizing ELBO or likelihood?
- This paper is proposing to 1. estimate the neighborhood distribution (instead of directly estimating the neighborhood) and 2. minimize Wasstertein distance as a core objective. Though the authors made an attempt to justify the design choice, I believe it could be compared as a sort of ablation study, which might strengthen the proposed method. --- I found Fig.6 and the first concern is resolved.
- I believe training time needs to be compared together in Fig. 7.
- Although performance improvement is shown by quantitaive experiments, it is not straightforward how the "context prediction" makes model performance better (qualitatively).

---

> ### Author Rebuttal · Authors · 2023-08-06
>
> *We thank Reviewer m5Qc for the positive review and valuable feedback. We are glad that the reviewer found that the motivation is agreeable, the writing is good, the method for motivation is reasonable, and experiments are done well. Please see below for our responses to your comments.*
>
> **Q1: Interpret “context prediction” term from the perspective of maximizing ELBO or likelihood.**
>
> A1: Connection between our context prediction loss in Eq.3 and ELBO:
>
> At time t:
>
> Loss of DDPM ( after reparamterization and scaling): $||x_0-\hat{x}_0(x_t,t)||_2^2$
>
> Loss of ConPreDiff:  $\sum\_{i = 1}^{x*y} [\mathcal{M}\_p(x\_0^{i} ,\hat{x}\_0^{i})+ \mathcal{M}\_n(\mathcal{H}\_{\mathcal{N}\_i},\hat{\mathcal{H}}\_{\mathcal{N}\_i})]$
>
> We let $\mathcal{M}_p,\mathcal{M}_n$ be square loss,
>   $\mathcal{M}_n(\mathcal{H}\_{\mathcal{N}\_i},\hat{\mathcal{H}}\_{\mathcal{N}\_i})=\sum\_{j\in \mathcal{N}\_i}(x\_0^{i,j}-\hat{x}\_0^{i,j})^2$,
>
>   where $x_0^{i,j}$ is the j-th neighbor in the context of $x_0^i$ and $\hat{x}_0^{i,j}$ is the prediction of $x_0^{i,j}$ from a denoising neural network. Following the notation in the main paper, we have
>
>   $\hat{x}_0^{i,j} = \psi_n(\psi_p(x_t,t)(i))(j)$,
>
>   where $\psi_p(x_t,t)(i)$ is the prediction of $x_0^i$ and $\psi_n$ is the neighborhood decoder. Compactly, we can write the denoising network (equation (1)) as:
>
>   $$\Psi(x_t,t)(i,j) =\left\\{ \begin{array}{l}\psi_n(\psi_p(x_t,t)(i))(j), & j \in \mathcal{N}\_i, \\\\ \psi_p(x_t,t)(i), & j=i \end{array} \right.$$
>
>   We can show that the DDPM loss is upper bounded by ConPreDiff loss, by reparameterizing $\hat{x}_0(x_t,t)$.
>   Specifically, for each unit i in the feature map, we predict the unit i and its neighbors using equation (1), and then we use the mean of  predicted value in the neighborhood as the final prediction:
>
>   $$\hat{x}\_0(x_t,t)(i) = 1/(|\mathcal{N}\_i|+1)*\sum\_{j \in \mathcal{N}\_{i}\ \cup\ \\{i\\}}\Psi(x\_t,t)(i,j)$$
>
>   Now we can show the connection between DDPM loss with the mean prediction and ConPreDiff loss:
>
> $$\begin{array}{rl}||x_0-\hat{x}\_0(x_t,t)||_2^2 & = \sum_i (x_0^i-\hat{x}\_0(x_t,t)(i))^2,\\\\ &=\sum_i (x_0^i-\sum\_{j \in \mathcal{N}_i\cup\{i\}}\Psi(x_t,t)(i,j)/(|\mathcal{N}_i|+1))^2,\\\\ &=\sum_i(\sum\_{j \in \mathcal{N}_i\cup\{i\}}(\Psi(x_t,t)(i,j)-x_0^i))^2/(|\mathcal{N}_i|+1)^2, \\\\ (Cauchy\ inequality) & \leq \sum_i \sum\_{j \in \mathcal{N}\_i\cup\{i\}} (\Psi(x_t,t)(i,j)-x_0^i)^2/(|\mathcal{N}_i|+1), \\\\ &=1/(|\mathcal{N}\_i|+1)\sum_i [(x_0^i- \psi_p(x_t,t)(i))^2+\sum\_{j \in \mathcal{N}\_i} (x_0^{i,j}-\hat{x}\_0^{i,j})^2] \end{array}$$
>
>
>   in the last equality, we assume that the feature is padded so that each unit i has the same number of neighbors $|\mathcal{N}|$.
>   As a result, the ConPreDiff loss is an upper bound of the negative log likelihood.
>
>
> **Q2: Training time comparison.**
>
> A2: We report the approximate time cost (seconds) of these models to train in a single step on the MS-COCO dataset in the table below. The additional context prediction head added to the models accounts for only a small portion of the parameters (0.13M), and the neighborhood distribution decoding of our method also does not incur significant additional training costs.
>
> |Training cost (secs/step) |LDM | DALL·E 2| Improved VQ-Diffusion| Imagen|
> | :-----| :----: | :----: |:----: |:----: |
> |Original Model  | 11.9 | 74.7 | 83.9 |198.3 |
> | + Context Prediction | 13.1 | 77.5 |87.6 |206.3 |
>
> **Q3: Qualitative comparison.**
>
> A3: More visualization results can be found in **global response pdf**, ConPreDiff better expresses local contexts and consistent semantics in generated images compared to other methods.
>
>
> **Q4: The meanings of "FNN" and “(0)” in line 138.**
>
> A4: We first mention FNN in L145 with feedforward neural networks, which contains two linear-BN-ReLU modules in our experiments. "(0)" is the superscript of $h_u$, which denotes the center point of neighborhoods $N^s_{i}$ (i.e., $h_i$).
>
> **Q5: Why not let FNN take target index as additional input and skip Eq. 7?**
>
> A5: Our distribution decoding loses original spatial orders of neighborhoods, and thus we use a **permutation invariant loss (Wasserstein distance)** for optimization. However, the Wasserstein distance between decoded neighborhood distribution and ground truth does not have a closed form. Thus we design **Eq.7 as an empirical surrogated loss of Wasserstein distance**. Taking target index as additional input may not well approximate Wasserstein distance.
>
> **Q6: What is the relationship between q in Eq.7 and stride s and K?**
>
> A6:
> 1. $K={(2s+1)}^2-1$
> 2. $q<K$
>
> **Q7: It is better to report the performance of directly estimating the neighbors as well.**
>
> A7: As you found in the Fig.6 of our paper, direct estimation of neighbors slightly improves the generation results but significantly incurs more training costs with larger strides. And our distribution decoding can achieve a better trade-off between FID score and training cost.
>
>
> **Q8: Why KL or JSD cannot be applied (L155)?**
>
> A8: Wasserstein distance can effectively measure structural similarity and impose structural constraint between the decoded distribution and the ground truth, via an optimal-transport loss. And Wasserstein distance can measure the similarity **between continuous and discrete distributions**, but KL or JSD can not well characterize the structural similarity between such distributions.
>
> **Q9: Are both terms in Eq. 3 used together for finetuning? What happens if Diffusion Models are trained from scratch with the proposed objectives?**
>
> A9: Both terms are used together. The convergence of our ConPreDiff would be quicker than the diffusion models that only use point-based reconstruction objective, because both terms play a cooperative role in maximizing ELBO as demonstrated in above responses.

---

> > ### Comment · Reviewer_m5Qc · 2023-08-21
> >
> > I keep my initial rating because my major concerns are resolved well.
> > Thank you for the rebuttal.

---

### Author Rebuttal · Authors · 2023-08-08

## Global Response
We sincerely thank all the reviewers for the thorough reviews and valuable feedback. We are glad to hear that the idea is interesting and well-motivated (all reviewers), the paper is well-written and easy to follow (Reviewer m5Qc, ydZz, and 2mXr), the proposed method is general and can be easily applied to various diffusion models (Reviewer DpwQ, ydZz and 2mXr), and performance improvement showed in experiments are promising (all reviewers).

We here summarize and highlight our responses to the reviewers:
* We make more visual comparisons to previous methods in the attachment pdf for more directly demonstrating the performance improvement (Reviewer m5Qc, DpwQ and ydZz) and better semantics expression (Reviewer 2mXr) of our proposed ConPreDiff.
* We provide additional theoretical derivation from the perspective of maximizing ELBO/likelihood (Reviewer m5Qc) to explain the cooperative relationships between our "context prediction" term and existing point-based denoising diffusion term (DDPMs), which is beneficial for model optimization and convergence (Reviewer qzoS and 2mXr).
* We also add some experiments to demonstrate the efficiency and soundness of the proposed "context prediction" (Reviewer m5Qc, DpwQ, and ydZz), and add more discussions about model limitations and societal impact (Reviewer DpwQ and ydZz, ethics reviews).

We reply to each reviewer's concerns in detail below their reviews. Please kindly check out them. Thank you and please feel free to ask any further question.

---

### Decision · Program_Chairs · 2023-09-21

**Decision:**

Accept (poster)

**Comment:**

This paper introduces a new method to improve diffusion-based image generation by predicting neighborhood context during generation without increasing inference cost. It outperforms standard diffusion models in various tasks and achieves state-of-the-art text-to-image generation on MSCOCO. Reviewers appreciate for the well-motivated idea, well-written paper and generalization for various diffusion models. The rebuttal also provides additional experiments and valuable insights to address reviewers' concerns very well. As a result, all reviewers have expressed scores above the acceptance threshold, and AC concurs with their recommendation to accept this paper. AC strongly encourages the authors to incorporate these valuable insights and constructive comments, possibly including discussion of potential negative applications and limitations, from the reviewers into the final version of the manuscript.